# Thermal Insulation Mattresses Based on Textile Waste and Recycled Plastic Waste Fibres, Integrating Natural Fibres of Vegetable or Animal Origin

**DOI:** 10.3390/ma15041348

**Published:** 2022-02-11

**Authors:** Andreea Hegyi, Horațiu Vermeșan, Adrian-Victor Lăzărescu, Cristian Petcu, Cezar Bulacu

**Affiliations:** 1NIRD URBAN-INCERC Cluj-Napoca Branch, 117 Calea Florești, 400524 Cluj-Napoca, Romania; andreea.hegyi@incerc-cluj.ro; 2Faculty of Materials and Environmental Engineering, Technical University of Cluj-Napoca, 103-105 Muncii Boulevard, 400641 Cluj-Napoca, Romania; 3IOSUD-UTCN Doctoral School, Technical University of Cluj-Napoca, 15 Daicoviciu Street, 400020 Cluj-Napoca, Romania; 4NIRD URBAN-INCERC Bucharest, 266 Șoseaua Pantelimon, 021652 Bucharest, Romania; 5MINET S.A., 12 Depozitelor Street, 240426 Ramnicu Valcea, Romania; cezar.bulacu@minet.ro; 6Faculty of Industrial Design and Business Management, “Gheorghe Asachi” Technical University of Iași, 29 Prof. Dr. Doc. Dimitrie Mangeron Street, 700050 Iasi, Romania

**Keywords:** thermal insulation, recycling, polyethylene terephthalate (PET), polyester (PES), natural fibres, sheep wool

## Abstract

The current context provides, worldwide, the need to identify solutions for the thermal efficiency of constructions, through sustainable and innovative methods and products. A viable solution is to produce thermal insulating products by carding-folding technology, using natural fibres and recycled polyethylene terephthalate (rPET) and polyester (rPES) waste, converted to fibres. This paper presents experimental results obtained after testing several thermal insulation composite products produced using a mix of sheep wool, cellulose, rPET and rPES fibres. The results of the research demonstrate the thermal insulation properties but, at the same time, identify the benefits of using such materials on the quality of the air in the interior space (the ability to adjust humidity and reduce the concentration of harmful substances). At the same time, the advantages of using sheep wool composite mattresses concerning their resistance to insect attack is demonstrated when compared with ordinary thermal insulation materials. Finally, sensitivity elements of these composites are observed in terms of sensitivity to mould, and to contact with water or soil, drawing future research directions in the development of this type of materials.

## 1. Introduction

According to the Energy Performance Directive (EPBD recast, 2010/31/EU) and the Energy Efficiency Directive (2012/27/EU) [1,2], nowadays all new buildings should have been designed according to the principles of the lowest possible energy consumption, and old buildings are currently designed thermally efficient as much as possible. One of the key factors in this design is the possibility of thermal insulation. On the other hand, the principles of the Sustainable Development concept and the need to implement the defining guidelines of the Circular Economy oblige us to identify new possibilities for the development of sustainable thermal insulation materials. If by 2020 it was estimated that buildings represented 30–40% of the global primary energy consumption and more than 25% of the sources of greenhouse gas emissions [3,4,5,6], the current pandemic context changed this paradigm. It has required the migration of activities in individual or smaller indoor spaces, with a lower degree of occupancy, so the adverse effects have increased substantially, leading to the increasing need to identify sustainable thermal envelope possibilities. Worldwide, various thermal insulation materials are currently used extensively, mainly expanded or extruded polystyrene or mineral wool. On the one hand, they offer satisfactory thermal performance but the impact of their production on the health of users and the environment cannot be neglected: most of the time, in addition to the polluting emissions that occur during the manufacturing process, these materials reduce the quality of the air in the living space by reducing the permeability of the walls to water vapour and various gaseous compounds. Consequently, there is an accumulation of indoor humidity leading to the development of moulds, the accumulation of volatile organic compounds or radon [7,8] and, finally, leading to the development of the so-called “sick building syndrome” [9,10,11,12,13,14,15]. Despite these disadvantages, thermal insulation based on expanded or extruded polystyrene, polyurethane foam, glass fibre or mineral wool accounts for about 87% of the market [16,17].

The use of plant and animal fibres and recycled waste in new bio-eco-innovative thermal insulation products, on the one hand, opens new possibilities for the development of such materials, but, on the other hand, brings new challenges, especially in terms of water and micro-organism behaviour [2,18]. The advantages of natural fibre-based thermal insulation materials are, above all, that they are relatively straightforward to obtain, from renewable resources, with low cost and environmental impact. Moreover, some characteristics that would appear to be disadvantages at first sight can easily be turned into benefits. For example, the feature of natural fibres to absorb moisture from the air can be seen as a benefit if viewed through the lens of sorption/desorption capacity, which will clearly contribute to the regulation of indoor air humidity [2,18,19,20,21,22]. A market share analysis [23] shows that growth in environmental awareness in the developed countries translates into the increased use of natural insulating materials, thus representing a growing market share in recent years. Correlated with market concerns, it is not surprising that research into the development of natural fibre-based thermal insulation materials has increased in scope over the last 20–25 years. Worldwide research has shown that insulation produced using natural plant or animal raw materials or recycled raw materials has lower greenhouse gas emissions than other insulation materials, e.g., mineral wool or polystyrene [24,25,26,27,28,29]. A comparative analysis of the impact of different types of insulation materials on global warming places sheep wool as having the lowest contribution (1.457 kg CO_2eq_/functional unit), 9–10 times lower than, e.g., extruded polystyrene (13.22 kg CO_2eq_/functional unit), and even half that of mineral wool insulation (2.77 kg CO_2eq_/functional unit) [30]. A similar report also ranked the lifetime energy consumption, from production to disposal and recycling, and in terms of stored energy (MJ/functional_eq_ unit)—much lower parameters for bio-eco-innovative thermal insulation [24,25,26,27,28,29,30]. Moreover, animal fibres, and especially sheep wool, due to the COOH-CHNH-R_2_ structure of the proteins in the wool yarn, contribute to the reduction in environmental pollution by adsorption on the yarn of gases such as SO_2_, NO_x_, aldehydes, or VOC, this adsorption being in some cases reversible (VOC, phenols) or irreversible (NO_x_, SO_x_, formaldehyde). In the case of sheep wool fibres, some research has shown that under conditions of 2.32 ppm sulphur dioxide concentration, the adsorption capacity of wool was initially 0.012 mg/g wool/min, reaching 0.003 mg/g wool/min after 60 min of exposure. Regarding the toxic products from cigarette smoke, a retention capacity of 40 mg cigarette smoke/g of wool was demonstrated. Moreover, other studies showed the capacity to reduce Cu^2+^ concentration by 41% in 1 h, and As^3+^ content by 34–53% in 1 h [31,32,33,34,35,36].

In terms of the mechanisms underlying these unique properties, natural fibres of animal and plant origin are defined by their structural specificity. Thus, in the case of wool yarns, the genetic structure, of a proteinaceous nature, with hydrophobic cuticles due to the presence of fatty acids covalently bound to the protein substrate, with a cortex composed of elongated cells containing more than 93% keratin (mass percentage) and 18 other α-amino acids and with a specific medullary channel, allows the absorption of 15–18% of the water vapour mass under normal conditions, but can reach up to 40% under saturated atmosphere conditions. The expansion of the medullary canal facilitates the absorption phenomenon, the diameter of the strand can increase by up to 18% and its length by 1%, and the formaldehyde neutralisation capacity is based on the keratin chemisorption reaction shown in Figure 1 [37,38,39,40]. In the case of fibres of plant origin, the absorption of atmospheric moisture is associated with the possibility of trapping hydroxyl (OH) groups and molecular water in the cell wall, which can be considered a cellulosic composite formed by crystalline microfibrils embedded in an amorphous lignin/hemicellulose/pectic matrix. Cellulose, consisting of linear chains of glucose grouped into units of microfibrils with high crystallinity, also has a paracrystalline component, favouring the binding of molecular water. Lignin is an amorphous cross-linked polymer composed of phenolic units; it has a lower (OH) group affinity compared with the polysaccharides characteristic of cellulose. Hemicellulose and the pectin component, predominantly amorphous polysaccharides, are very accessible for the addition of water molecules [22].

Unlike fibres of plant or animal nature, those from recycled waste plastics such as polyethylene terephthalate (PET) or polyester textile waste (PES) are relatively inert in terms of their chemical binding capacity to water or other compounds. Nonetheless, they offer different advantages, and the need to recycle and reuse them is imperative under the Waste Directive 2008/98/EC [41]. Their recycling successfully addresses the need to reduce the quantity of polymer waste and to obtain new products with reduced emissions and the sustainable use of non-renewable raw material and energy reserves. Polyethylene terephthalate (PET) is a material of great interest in the materials sector in general and, by extension, in the building materials sector as it is 100% recyclable, the only more easily recyclable material being aluminium. The cost of recycled polyethylene terephthalate (R-PET) is 20–60% lower than the cost of polyethylene terephthalate produced from virgin raw material (V-PET). Additionally, the reduction in energy consumption is 50–70% and the reduction in petroleum feedstock consumption is 50–60% [42,43,44,45,46,47]. Worldwide, the main use of R-PET is for fibre production, accounting for more than 70% of all polyethylene terephthalate fibres produced [29,42,44]. Studies have demonstrated the possibility of using polymeric fibres from recycled materials to produce non-woven thermal insulation materials for the construction sector, thus identifying the potential for conserving natural resources and reducing environmental impact. In the case of polyethylene terephthalate (PET) mattresses, thermal conductivity varies with increasing density. For the same thickness of non-woven thermal insulation product, a low density ensures a high thermal insulation potential, but for mechanical strength reasons this density must be increased. A balancing calculation of thermal and mechanical performance is necessary, also taking into account that around a density value of 60 kg/m^3^, the phenomenon of heat transfer through conduction becomes dominant and thermal insulation performance decreases significantly [48,49,50,51]. Research has demonstrated the possibility of obtaining thermal insulation mattresses produced using R-PET fibres characterised by a thermal conductivity coefficient around values of 0.035 W/mK [29]. Based on the research, the quantifiable indicator TIV “Thermal Insulating Value” was identified, which indicates that the thermal insulating efficiency of a product depends on the thermal conductivity coefficient of the material, the thickness of the product and the thermal emission of the surface. For thermal insulation products produced using PET, the TIV coefficient ranges from 41.21% to 52.15% for thicknesses in the range 3.54–7.97 mm, increasing alongside increasing thickness [48,49,50,51]. At the same time, the advantage of this type of fibre is its reduced sensitivity to water and increased resistance to mould action.

Approximately 5.8 million tons of textile waste are produced annually in the EU, of which only 25% is recycled and reused, although the recycling potential is 95% [52,53,54]. One of the possibilities for recycling is the transformation into fibres which, although not fully suitable for reuse in the textile industry, can successfully become a raw material in the non-woven thermal insulation sector for the construction industry. Research results to date [55,56] have shown that recycled textiles have competitive thermal properties, thermal conductivity and thermal diffusivity, and can be used as an alternative to commercial thermal insulation materials (extruded polystyrene or mineral wool) in the construction sector. Danihelová et al. [55], in agreement with other reports [48] showed that recycled textile waste mattresses can be good thermal insulators, characterised by a thermal conductivity coefficient around 0.033 W/mK.

Therefore, producing thermal insulation mattresses from a mixture of recycled PET fibres, recycled PES, fibres of plant origin and fibres of animal origin makes it possible to exploit the basic principle of composite materials: the accumulation of benefits.

This paper aims to present the possibility of producing composite materials, of the non-woven type (mattresses), intended for use as thermal insulation materials in the construction sector, and through the experimental results obtained, to identify the benefits that these products can have on the air quality of the indoor space and, implicitly, on the health of the users, also identifying possible weaknesses to draw future research directions.

## 2. Materials and Methods

### 2.1. Materials

The experimental testing was carried out for six types of non-woven thermal insulation materials (thermal insulation mattresses), presented in Table 1, identified with codes P1–P6. They were produced by carding-folding technology with thermal consolidation at low temperature (110–180 °C), from a mixture of raw materials. The mixture of fibres used to produce the six heat-insulating mattresses (mass ratio) is characterised by three parts combed sheep wool; 1.5 parts siliconised cellulosic fibre; 2.5 parts heat-sealable bicomponent recycled polyethylene terephthalate (PET) fibre; and 3 parts recycled polyester fibre (PES). Sheep wool yarn was obtained by splitting combed sheep wool, which was purchased in bulk from local traders. The siliconised cellulosic fibre is characterised by an average linear density of 2 dtex and length of 20 mm. Recycled PES fibre is characterised by an average linear density of 7 dtex and a length of 65 mm. Recycled PET fibre is bicomponent, max. 9 dtex and max. 80 mm length.

The six thermal insulation mattresses, although produced using the same fibre blend, had different bulk densities and thicknesses, characteristics that were controlled by controlling specific parameters on the technological flow. The production of the materials took place in an industrial system, on the technological flow of MINET S.A. Rm.—Vâlcea, Romania.

Given the intended field of use, several categories of measurable indicators have been assessed for each type of thermal insulation mattress according to EAD 040456-00-1201/2017 “In situ formed loose fill thermal and/or acoustic insulation material made of animal fibres” and EAD 040005-001201/2015 “Factory-made thermal and/or acoustic insulation products made of vegetable or animal fibres”. These indicators are: physical–mechanical (heat transfer coefficient, compressive strength, tensile strength, elongation failure, short-term water absorption) and durability, biological resistance and impact on human health (sorption–desorption characteristics of atmospheric moisture, water vapour permeability, resistance to the action of micro-organisms, formaldehyde neutralisation capacity and corrosivity).

### 2.2. Thermal Insulation Performance

Heat transfer coefficient, λ_10,ct._ (W/mK) was determined using a FOX 314 conductivity meter by the hot plate method at a temperature difference between the plates of 10 °C. Samples were placed between the two plates, and then a temperature gradient was established over the probe thickness. The equipment automatically detects equilibrium conditions and determines the heat transfer resistance R (m^2^K/W) and, for the homogenous probes, the equivalent heat transfer coefficient.

### 2.3. Mechanical Performance

In order to evaluate the compressive strength (universal testing machine, type ZDM-5/91, VEB Leipzig, Germany), a compressive load was applied with uniformly increasing speed, perpendicularly and uniformly distributed over the surface of the specimen, and the degree of indentation of the specimen was recorded. The compressive strength was calculated as the ratio of the applied force to the area of application, for an indentation of 10% of the initial thickness of the specimen.

Tensile behaviour was assessed by recording the maximum tensile strength and elongation at failure of specimens taken in the longitudinal/transverse direction of the material (Shenzhen Wance Testing Machine Co., Ltd., Shenzhen, China). According to the standards in force, short-term water absorption is an indicator of the material behaviour in the situation caused by a 24 h rain period during the construction process. The test specimen was placed with the bottom side in contact with water for 24 h and its mass change in relation to the exposed surface area was measured.

### 2.4. Behaviour in Water and in the Presence of Water Vapour

Water vapour sorption/desorption capacity was quantified by plotting characteristic curves of specimen mass variation against relative air humidity. Thermal-insulation conditions, 23 °C, controlled by relative air humidity (RH) of 30%, 45%, 60%, 80%, and 95%, were chosen for testing.

Water vapour permeability was evaluated using test specimens sealed over circular cups in which water vapour pressure was kept constant by a saturated potassium nitrate solution under specified conditions (23 ± 2 °C). Moisture transfer was determined as the change in weight of the test system when constant vapour flow was reached.

The samples were constantly weighted using an analytical balance from Kern & Sohn GmbH, Albstadt, Germany, with a precision of 0.0001 g.

### 2.5. Resistance to the Action of Biological Agents

Mould resistance was determined by exposing the specimens for a defined period of time (4 weeks), at a constant temperature of 23 ± 2 °C, in an environment with high humidity, after which a visual and microscopic evaluation was carried out on the appearance of signs of growth of microorganisms, after which the product was given a certain class indicated in Table 2 and Table 3, respectively.

Biological resistance to worm and insect attack was assessed by evaluating the viability of the first generation and the development of the second generation of insects, the common moth (*Order lepidoptera*), under conditions of the contact of specimens with eggs, larvae and adult insects for 6 months. The classification into resistance classes by visual assessment of the damage caused by was carried out according to Table 3.

Resistance to soil biological attack was assessed by microscopic analysis to identify signs of microorganism and worm growth and by determining the percentage reduction in tensile strength and elongation at break under conditions of complete burial of the specimens in soil with a water holding capacity of 60% for 4 weeks and 6 months, respectively.

The analysis of the samples in terms of resistance to the action of biological agents was carried out using a Leica DMC2900 microscope (Leica Microsystems GmbH, Wetzlar, Germany) and results/images were obtained using the dedicated image capturing application (Leica Application Suite, Leica Microsystems GmbH, Wetzlar, Germany).

### 2.6. Metal Corrosion Development Capacity

The metal corrosion development capability provided information on the level of corrosivity of the thermal insulation material on Cu/Zn metal elements (metal parts that may come into contact with the thermal insulation material in place) by allowing the migration of corrosive agents adjacent to the thermal insulation material towards these metal elements. The ability of metal corrosion to develop was assessed by visual and microscopic evaluation of the appearance of signs of corrosion on the surface of metal plates (Cu sheet or galvanised steel sheet) after they had been kept in contact with the surface of specimens of thermal insulation material saturated with water, for 4 weeks. The analysis of the samples in terms of metal corrosion development capacity was carried out using the same equipments as the ones used for the analysis of the action of biological agents.

### 2.7. Impact on Indoor Air Quality

The formaldehyde neutralisation capacity was assessed by recording the variation in formaldehyde concentration in a sealed enclosure of which the base was completely covered with a heat-insulating mattress. Initially, the atmosphere in the 0.1 m^3^ volume enclosure was enriched with formaldehyde to a concentration of more than 25,000 μg/m^3^, equivalent in terms of health impact to a “toxic level with neurotoxic manifestations”. Formaldehyde concentration monitoring was performed using commercial equipment equipped with a formaldehyde sensor and electronic display (HCHO & TVOC Meter, Shenzhen Everbest Machinery Industry Co., Ltd., Shenzhen, China). When the formaldehyde concentration in the monitored volume fell below the limit level of 200 μg/m^3^, which is equivalent in terms of the health impact on the population to a state of “no signs of discomfort or irritation”, a new amount of formaldehyde was introduced into the test enclosure to assess the maintenance of the formaldehyde-neutralising capacity of the heat-insulating mattress over time. The specimen used, with dimensions equal to the base dimensions of the enclosure, 600 × 450 mm, was covered on the lower face and on the edges with aluminium foil, so that formaldehyde transfer was achieved only through the upper, exposed face.

To ensure repeatability, five determinations were performed for each test, the results being presented as the mean values of the individual values.

The results obtained experimentally were analysed both in comparison with other references in the literature and with existing data in the NIRD URBAN-INCERC databases on the performance of similar thermal insulation products produced by other Romanian manufacturers.

## 3. Results and Discussions

As is well known, some of the performances analysed are influenced by the material, by its composition, not by its thickness or density. Following the experimental tests carried out, several experimental results were recorded; some expected, others less anticipated. Thus, in terms of thermal insulation performance, the products tested meet the specific requirements of the intended field of use, and the results are in accordance with the specifications in the literature in terms of the heat transfer coefficient, the values recorded being in the ranges 0.034–0.042 W/mK for products tested with their full thickness and 0.030–0.034 W/mK for products tested at half thickness by tamping (Table 4), giving heat transfer resistances, R, in the range 0.55–1.46 mK/W^2^ for products tested with their full thickness and 0.33–0.89 mK/W^2^ for products tested at halved thickness. Comparing the experimental results obtained with the declared performance of non-woven thermal insulation products made exclusively or predominantly from sheep wool by local producers (Figure 1), it can be said that, in terms of thermal insulation performance, the analysed composite products fall within the general trend, in the higher performance area, with heat transfer coefficient values of 0.032–0.050 W/mK for the comparison products. It should be specified that the comparison values presented in Figure 2 are the results of experimental tests carried out at NIRD URBAN-INCERC Cluj-Napoca Branch Laboratory, on some insulation products present on the local market, with data collected in the past five years. As seen in Figure 1, in terms of thermal insulation performance, analysed on the basis of the measurable indicator λ—heat transfer coefficient, it can be said with certainty that the tested specimens P1–P6 correspond to the intended field of use. The values recorded are comparable with those presented by various types of common thermal insulation materials (expanded polystyrene, extruded polystyrene, mineral wool, etc.) as well as alternative products such as mattresses produced only with sheep wool. It should be noted, however, that under current legislation in the field, this validation in terms of thermal insulation performance is insufficient, and at least a complement is needed in terms of operational safety, durability, and analysis, with regard to environmental and user health impacts.

The density of the material (Table 4) has a significant impact on the thermal conductivity, market acceptance and feasibility when put into operation. When plotted against the density, thermal conductivity first drops as the air cavities between fibres are reduced in volume when increasing in density. An optimum value is reached, the heat transfer through conduction grows as the contact between fibres increases, and the convective part of heat transfer cannot compensate anymore. Generally, a mattress with a higher density (around 30 kg/m^3^) has better compressive strength values and is easier to deal with from the workers’ perspective. However, we have to keep in mind that market acceptance is of the essence. Therefore, production prices should be kept as low as possible. One should observe and balance all these aspects for a viable product, and material density is the common denominator of all these aspects.

Additionally, in the same vein, as presented in Table 4, it is not possible to establish a pattern regarding the values of density vs. thickness vs. thermal performance, since for each case there are advantages and disadvantages, thermal insulation performance being influenced by factors other than dimensional ones.

In terms of mechanical strength, as expected due to the non-woven mattress technology used, low values were recorded. Thus, the compressive strengths for an indentation of 10% of the original thickness (Table 4) was in the range 0.10–0.20 kPa. The tensile strength was in the range 0.05–0.13 N/mm^2^ for the specimens cut in the longitudinal direction of the thermal insulation mattress, with a total elongation at break of 22.86–47.60%, and in the range 0.03–0.19 N/mm^2^, with a total elongation at break of 28.64–43.45% for the specimens cut in the transverse direction.

The short-term water absorption recorded in the range of 0.07–0.38 kg/m^2^ confirmed the sensitivity of this type of thermal insulation material to water and the need for protection both during storage, transport, and commissioning and during the entire service life. However, compared with similar products produced under similar conditions, but exclusively from natural fibres of animal origin (sheep wool), an improvement in the parameter is noted by reducing it even 10 times. This improvement was due to the contribution of the other types of fibres introduced in the raw material mix, in particular recycled PET fibres.

In terms of water vapour sorption/desorption capacity (Figure 3), it is noticeable that the potential regulating character of the indoor air humidity, and therefore its quality, is preserved, a characteristic of similar sheep wool products. The placement of the sorption curve, in all cases, below the desorption curve, means that, although permeable to water vapour, these innovative thermal insulation mattresses, under conditions of increased humidity of the ambient environment, are able to retain a certain amount of water, the results being in agreement with those reported in the literature [2,18,19,20,21,22,25]. As the ambient humidity is reduced, this water will be released to the environment, but at a lower rate than the retention rate, as demonstrated by the slope of the sorption-desorption curves. As a result of this behaviour, it can be seen that this type of product contributes not only to the thermal insulation of the interior space but also to the regulation of its humidity. By analysing the shape of the sorption curves, it can be appreciated that they are in accordance with the specifications of the literature [2,18,19,20,21,22,25], which indicate that water absorption of up to 5% of the dry weight of the fibre takes place at a higher speed, with water binding strongly, after which the phenomenon is influenced by the relative humidity of the environment, the speed stabilising towards the saturation zone, which is also identified by analysing the slope of the sorption–desorption curves. The identification of a slope change zone on each of the desorption curves was considered to delimit a range of relative air humidity in which the activity of regulating atmospheric humidity by releasing water is characterised by a maximum yield. Mathematical modelling of the air moisture sorption/desorption process revealed first-order equations (ax + b), i.e., that these phenomena are influenced by relative air humidity.

Satisfactory results were also obtained in terms of water vapour permeability. These thermal insulation composites have a good water vapour permeability, which will contribute substantially to the air quality of the indoor space, allowing the transfer of moisture from the inside to the outside and thus making a significant contribution to reducing the risk of mould growth or the development of other types of micro-organisms. Experimental testing showed the following characteristic parameters: water vapour permeance, W, in the range 8.83–18.43 m/mhPa^2^; vapour resistance, Z, in the range 0.06–0.11 mPa/mg^2^; water vapour permeability, δ, in the range 0.31–0.75 mg/mhPa; water vapour diffusion resistance factor, μ, in the range 0.95–2.32; and water vapour diffusion equivalent air layer thickness, s_d_, in the range 0.04–0.08 m. Compared with other thermal insulation products, the water vapour diffusion resistance factor, μ, of the tested thermal insulation composite products is similar to that of other non-woven thermal insulation products (sheep wool mattresses) and clearly superior to commonly used thermal insulation products such as mineral wool and, even more so, expanded or extruded polystyrene. In fact, this is one of the great advantages of the composite thermal insulation mattresses tested.

Microscopic analysis of the specimens after testing them under mould growth conditions showed numerous areas where mould develops (Figure 4). However, it was not possible to identify precisely which type of fibre of the four raw materials used was favoured, and to what extent it contributed to mould growth. After the first four weeks of testing, there were already clear signs of mould, visible under the microscope, on the whole surface of the specimen and in its depth, the classification according to Table 2 being class 1 (growth invisible to the naked eye, but clearly visible under the microscope), category 2–3 (material does not resist fungal attack; it contains nutrients that allow micro-organisms to grow). Continued testing over the next 4, 8, 16 weeks revealed continued mould growth over time, which indicated an urgent need to identify a treatment solution to reduce mould susceptibility.

Biological resistance to attack by worms and insects has highlighted an advantage of these thermal insulation materials compared with similar thermal insulation made exclusively or mainly of sheep wool: visual and, subsequently, microscopic analysis (Figure 5) showed that they fall into category 1A (Table 3) and that the bio-eco-innovative products tested do not represent a viable environment for moths (*Order lepidoptera*), their eggs and larvae. After 7 days of exposure, all adult insects died and there was no evidence of egg laying for the further life cycle. In the case of samples on which moth eggs and larvae were artificially deposited, life cycle interruption was recorded and they did not develop into larvae or adult insects, respectively. Testing continued by follow-up for 56 days with no signs of insect development or the appearance of developmental signs recorded. In contrast to thermal insulation mattresses made exclusively or predominantly of sheep wool, the risk of moth colonies developing, characteristic of products made of wool or various other textile fibres, is minimised in this case. Although clear and proven identification has not been achieved, it is assumed that the introduction of recycled PET and vegetable fibres and their siliconisation has contributed to the insecticidal action of the tested material.

In terms of resistance to ground action (Figure 6 and Figure 7), after keeping the specimens buried in soil for one month, there were already signs of degradation. Without showing the exact degradation at the level of the fibres, deposits of microorganisms, mould (Figure 6b,e–h), worm colonies, worm eggs (Figure 6a,c,d,g,h), etc., can be observed on the fibres, which leads to the hypothesis that this material is a favourable environment for the development of microorganisms and worms. After six months in the soil, signs of the degradation of the specimens are evident, as well as indications of the development of colonies of microorganisms (Figure 7a,b,e) and worms (Figure 7c–f); the presence of at least two types of moulds (green and black—Figure 7a,b,e), and the presence of at least two types of small organisms, worms (Figure 6e,f), their larvae and eggs (Figure 7c,d) are identified. These results, on the one hand, indicate the need to impose restrictions in terms of use so as not to come into direct contact with the soil, but, on the other hand, they represent the fact that, after the end of the products’ life span, they become waste, which could be directed towards the realisation of composite matrices that, in combination with the soil and specific nutrient substrates, allow the development of substrates favourable to plant cultivation; a trend reported in the literature [57,58,59,60,61,62].

In terms of the measurable parameters evaluated, tensile strength and elongation at break (Figure 8), it can be observed that after one month of keeping the specimens buried in the soil, the tensile strength was little influenced, the loss of strength being at most 1 N/mm^2^, which could indicate the maintenance of the thermo-welding effect, but the elongation at break was reduced by 7–30%, depending on the tested composite, which could indicate a degradation in the fibres, becoming more prone to breakage and having less elastic. After six months in the soil, the specimens were more strongly degraded, with measurable parameters indicating a reduction of at least 20% in tensile strength and elongation at break, which could be considered a sign of the degradation of both the degree of thermal welding between fibres and the integrity and performance of the fibres in the matrix of the thermal insulation material.

As can be seen in Figure 9 and Figure 10, in humid conditions, when in contact with the composite thermal insulation mattresses, the Cu or galvanised metal elements undergo slight corrosion, manifested by the loss of gloss and the formation of a small quantity of corrosion products, which are easily removed by washing. It can therefore be said that, in terms of the migration of corrosive agents adjacent to the thermal insulation material towards these metal elements, bio-eco-innovative thermal insulation products do not provide corrosion protection, as atmospheric oxygen, water and other corrosive agents easily reach the metal surface, nor do they accelerate this process.

The formaldehyde neutralisation capacity (Figure 11) revealed the good ability of the composite thermal insulation matrix in terms of formaldehyde neutralisation. In the first 72 h after the start of the test, the composite matrix succeeded in neutralising a sufficient amount of formaldehyde, to reach, from a concentration in the zone of high risk, high discomfort, headaches, possible skin allergies, sickness and nausea, a zone of moderate risk concentration, and in the following hours this concentration zone is also exceeded, entering a zone of formaldehyde concentration in the air that does not pose a risk to the population. After neutralisation of the first dose of formaldehyde, the replenishment of the air with the chemical agent does not pose a problem for the test product to develop a new neutralisation reaction. In both cases, the neutralisation of formaldehyde occurs at a high rate initially in the high concentration zone, and then decreases as the concentration of formaldehyde in the air decreases. It should be noted, however, that under normal conditions of use in the living space, it is rare that the formaldehyde concentration reaches concentrations similar to those at which the test started. These concentrations have been chosen precisely to evaluate the behaviour of the product under extreme conditions. It can therefore be said that the composite matrix obtained by recycling and reusing PET and PES waste, together with the use of natural vegetable fibres and sheep wool, successfully copes with and certainly contributes to improving indoor air quality, in fact maintaining the special character of sheep’s wool, which, through its keratin structure, is the neutralising agent. Moreover, according to the literature, formaldehyde adsorbed by chemisorption mechanisms, inducing chemical, stable bonds, also has the advantage that it becomes, indirectly but effectively, a passive treatment to increase resistance to attack by microorganisms, insects and their larvae [31,37].

Analysing this neutralisation rate in relation to the momentary concentration, and relating it to the time interval between two consecutive measurements, it can be seen that, as the concentration of formaldehyde in the indoor air decreases, the amount of formaldehyde neutralised per unit time decreases.

## 4. Conclusions

This work aimed to analyse the possibility and benefits of making composite products based on PET, recycled PES and natural fibres of plant and animal origin, of non-woven type (mattresses), intended for the thermal insulation of buildings, also highlighting the weaknesses, identified as possibilities for improvement through the development of future research.

By design, the analysed thermal insulation products identify clear possibilities to implement the guidelines of the Circular Economy concept, contributing both to waste recycling and to sustainable development through the use of components from renewable sources, with the final result of reducing energy consumption and increasing air quality in the living space.

Therefore, the originality of this work is a result of the approach to the topic of insulating products, combining the need to implement the principles of the Circular Economy with existing knowledge to date, but limited to the development of alternative insulating products using one or two of the four raw materials used in this research. The article focuses on an original formula for a thermal-insulating product which incorporates 55% recycled materials (25% rPET and 30% rPES), 30% wool by-products or virgin wool with low-quality fibre impossible to use in the textile industry, and 15% siliconised cellulosic fibre. Additionally, in general, the literature is mainly concerned with the analysis of thermal performance and less with compatibility and the influence on indoor environmental quality and user health.

Although in terms of mechanical resistance, these products have low performance, requiring the imposition of limiting conditions of use (mounting in wooden frames, protection of exposed surfaces, protection from prolonged contact with water or soil), they have a number of other significant advantages compared with similar products made of a single type of fibre:The thermal insulation performance, evaluated by the two indicators, thermal conductivity coefficient and heat transfer resistance, indicates a good thermal insulation capacity, compared with some thermal insulation products currently used on the construction market and, therefore, a significant contribution to the reduction in energy consumption used to ensure thermal comfort. If the thickness is reduced by half, as a result of an impact load, for the same type of sample, a decrease in the heat transfer coefficient can be observed and therefore an increase in thermal insulation performance by 15.7% (P1), 14.2% (P2), 19.8% (P3), 13.8% (P4), 18.2% (P5) and 10.0% (P6);The water absorption, although having values that impose the need for protection in contact with water, is reduced to values up to 10 times lower than the values characteristic of sheep wool mattresses. It is estimated that the improvement of this parameter is due to the inclusion in the composite matrix of recycled PET and vegetable fibre components;The composite material retained the specific capacity of natural fibres to sorb/desorb atmospheric humidity, thus contributing to the improvement of air quality in the indoor space. Additionally, the component of animal origin, sheep wool, in the composite matrix, is sufficiently efficient also in terms of formaldehyde neutralisation capacity, thus making another significant contribution to improving indoor air quality;The evident water vapour permeability characteristic of the thermal insulation products analysed demonstrates, once again, their benefits in ensuring adequate indoor air quality, contributing significantly to ensuring the breathability of the walls and thus reducing the risk of moisture accumulation and mould growth;The recycled waste components, and probably the siliconised component of the fibrous matrix, significantly improved resistance to insect action. This improvement is particularly important as it is known that thermo-sheep wool mattresses are generally very sensitive to insect attack, especially moths;From the point of view of the corrosivity on metallic elements with which these products could come into contact, the experimental results obtained allow the assessment that the composite thermal insulation matrix does not contribute to provide corrosion protection, but neither does it accelerate the corrosion phenomenon;The reduced resistance to mould attack and contact with the soil, although they could be seen as disadvantages of these thermal insulation materials, are a challenge for the development of future research directions, but, above all, they can be seen as a first step towards the development of new possibilities for waste recycling by developing materials that can be successfully used in the agricultural field, such as growing substrate or protective material.

Therefore, based on the experimental results and in accordance with references in the literature, it can be said that thermal insulation mattresses made of recycled PET and PES fibres, natural plant fibres and sheep wool can be an alternative to commonly used thermal insulation products (expanded polystyrene, extruded polystyrene, mineral wool, etc.), and, at the same time, can contribute to increasing the quality of life of the population by ensuring better indoor air quality.

## Figures and Tables

**Figure 1 materials-15-01348-f001:**
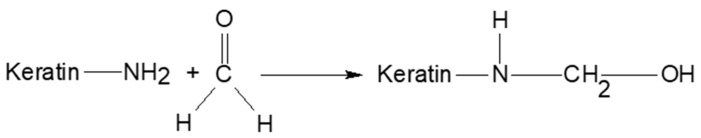
Keratin chemisorption reaction.

**Figure 2 materials-15-01348-f002:**
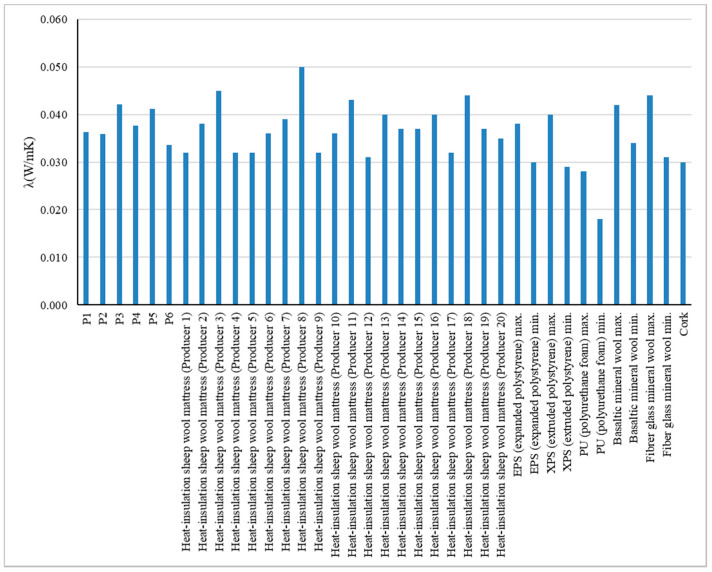
Thermal conductivity coefficient recorded for samples P1–P6 in comparison with the thermal conductivity coefficient of sheep wool mattresses and common thermal insulation materials available on the building materials market.

**Figure 3 materials-15-01348-f003:**
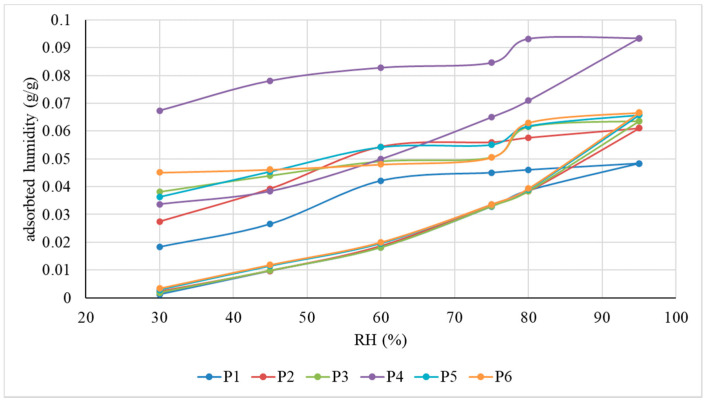
Atmospheric moisture sorption–desorption curves.

**Figure 4 materials-15-01348-f004:**
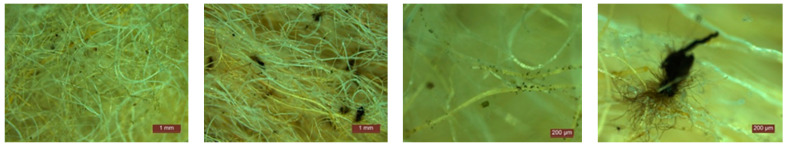
Image captures obtained by microscopic analysis of test specimens for mould resistance.

**Figure 5 materials-15-01348-f005:**
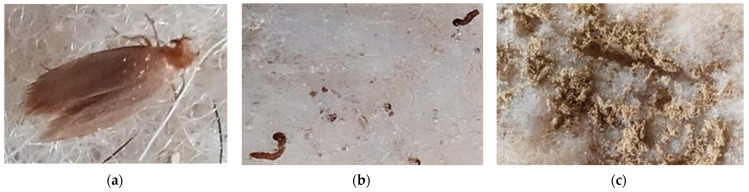
Exposure of heat-insulating mattresses specimens to assess biological resistance to attack by worms and insects: (**a**) in the presence of adult insects; (**b**) in the presence of larvae; (**c**) in the presence of insect eggs.

**Figure 6 materials-15-01348-f006:**
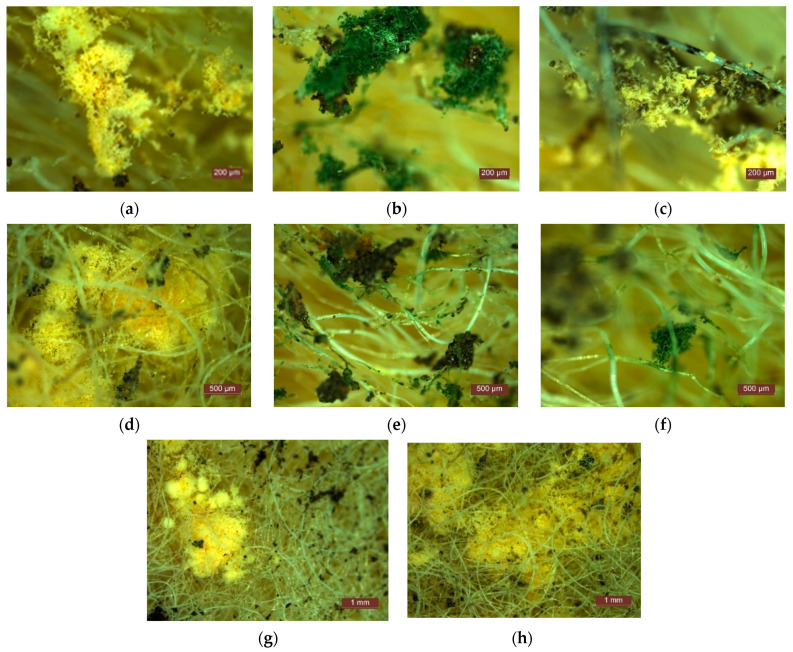
Instances regarding the microscopic analysis of test specimens by holding in soil for one month (**a**–**h**).

**Figure 7 materials-15-01348-f007:**
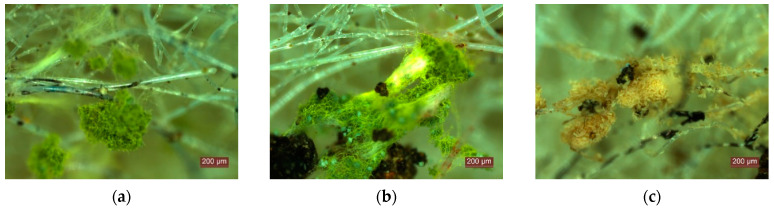
Instances regarding microscopic analysis of test specimens by holding in soil for six months (**a**–**f**).

**Figure 8 materials-15-01348-f008:**
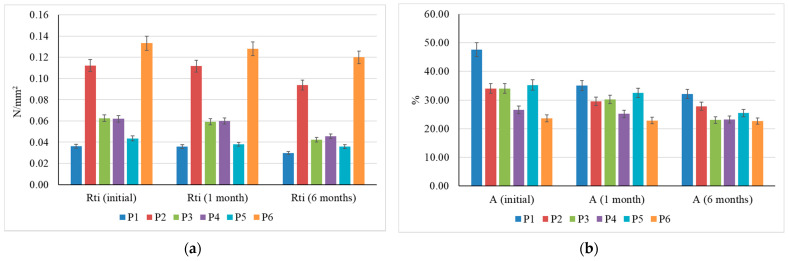
Variation in tensile strength (**a**) and elongation at break (**b**) after soil retention.

**Figure 9 materials-15-01348-f009:**

Evolution of the appearance of metallic Cu: (**a**) initial appearance of the metal specimen; (**b**) appearance of the metal specimen after testing; (**c**) appearance of the metal specimen after testing and washing of corrosion products.

**Figure 10 materials-15-01348-f010:**
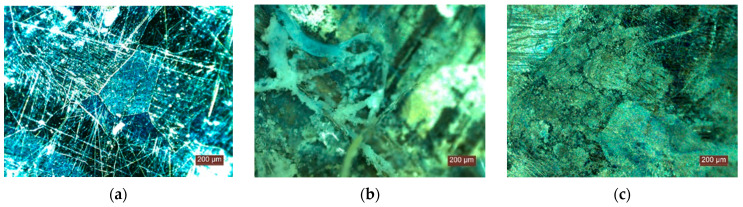
Evolution of the appearance of galvanised steel specimens: (**a**) initial appearance of the metal specimen; (**b**) appearance of the metal specimen after testing; (**c**) appearance of the metal specimen after testing and washing of corrosion products.

**Figure 11 materials-15-01348-f011:**
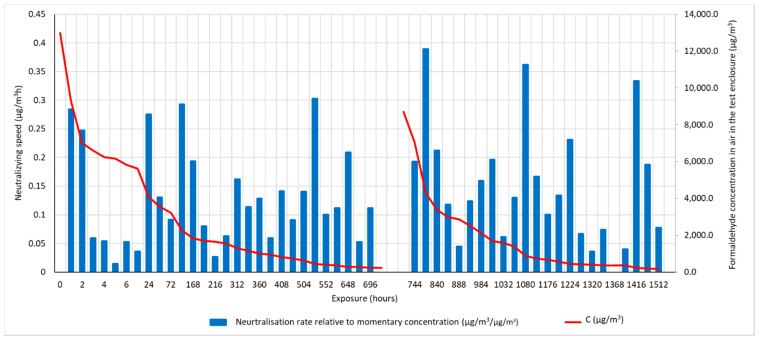
Variation over time in formaldehyde concentration in the air in the room where the heat-insulating mattress was placed and its neutralisation rate (neutralisation rate relative to the momentary concentration represents the amount of formaldehyde neutralised from time t of measurement to time t + 1 of measurement, relative to the formaldehyde concentration existing at time t).

**Table 1 materials-15-01348-t001:** Thermal-insulation mattresses produced using textile waste, recycled PET waste and natural fibres of plant and animal origin.

Identification	Thickness (mm)	Apparent Density (kg/m^3^)
P1	19.99	18.56
P2	29.99	29.22
P3	49.98	21.41
P4	49.98	25.23
P5	59.99	16.38
P6	29.99	39.99

**Table 2 materials-15-01348-t002:** Criteria for assessing mould resistance.

Fungal Growth Class	Evaluation of Fungal Growth	Product Category	Product Performance
Class 0	No sign of growth under the microscope.	Category 0	The material is not a nutrient medium for microorganisms (it is inert or fungistatic)
Class 1	Growth invisible to the naked eye but clearly visible under a microscope.		
Class 2	Increase visible to the naked eye, covering up to 25% of the test area.	Category 1	The material contains nutrients or is so poorly contaminated that it allows very little growth
Class 3	Visible growth to the naked eye, covering up to 50% of the test area.		
Class 4	Considerable growth, covering more than 50% of the test area.	Category 2–3	The material is not resistant to fungal attack; it contains nutrients that allow micro-organisms to grow.
Class 5	Strong growth, covering the entire test area		

**Table 3 materials-15-01348-t003:** Criteria for assessing insect resistance.

Estimation of Surface Degradation		Hole Estimation	
Symbol	Ruptures-Visibly Degraded Surface	Symbol	
1	No visible damage	A	No visible holes
2	Reduced degradation	B	Partially destroyed yarns and fibres
3	Moderate damage	C	Reduced number of holes; destroyed yarns and fibres
4	Major damage	D	Large holes

**Table 4 materials-15-01348-t004:** Physical-mechanical performance of thermal-insulation mattresses produced using recycled textile and PET waste and natural fibres of plant and animal origin.

Identification Code		P1	P2	P3	P4	P5	P6
Heat transfer coefficient at full thickness (W/mK)		0.0362	0.0358	0.0421	0.0377	0.0412	0.0336
Heat transfer coefficient at 1/2 of thickness (W/mK)		0.0306	0.0308	0.0338	0.0325	0.0337	0.0303
Heat transfer resistance at full thickness (mK/W^2^)		0.55	0.84	1.19	1.33	1.46	0.89
Heat transfer resistance at 1/2 of thickness (mK/W^2^)		0.33	0.49	0.74	0.77	0.89	0.50
Compressive strength (kPa)		0.17	0.18	0.14	0.20	0.10	0.11
Tensile strength (N/mm^2^)	longitudinal direction	0.05	0.11	0.06	0.06	0.04	0.13
	transverse direction	0.03	0.11	0.05	0.08	0.05	0.19
Elongation at break (%)	longitudinal direction	47.60	34.02	34.03	26.62	35.27	22.86
	transverse direction	43.45	37.71	28.64	38.26	31.82	31.24
Short-term water absorption (kg/m^2^)		0.069	0.223	0.132	0.121	0.387	0.350
Water vapour resistance (m∙h∙Pa/mg^2^)		0.06	0.06	0.09	0.11	0.09	0.08
Water vapour resistance factor, µ		1.35	1.41	1.26	1.61	0.95	2.32

## Data Availability

The data presented in this study are available on request from the corresponding authors.

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
