# Peer review of "Thermal Insulation Mattresses Based on Textile Waste and Recycled Plastic Waste Fibres, Integrating Natural Fibres of Vegetable or Animal Origin"

_materials, 2022, doi:10.3390/ma15041348_

Round 1

Reviewer 1 Report

General:

The topic “Eco-Friendly Alternative to Classic Thermal Insulation Products - Thermal Insulation Mattresses Based on Textile Waste and Recycled Plastic Waste Integrating Natural Fibers of Plant and Animal Origin” is interesting, relevant, and presented results are a good attempt to contribute to advancing the state of the art. However, the paper needs major revision before it is accepted for publication in Materials, MDPI.

Comments:

  1. Title of the manuscript is way too long. I would suggest making it concise.
  2. Line 20-24, The sentence is too long. Kindly divide it into two sentences.
  3. Line 25, It would be better if authors can provide exact four types/origin of the fibers.
  4. The novelty of work is not clear. I would recommend authors should clearly mention the novelty in their work.
  5. Kindly mention the specific reason for different apparent densities of the mattresses. Also please clearly state the purpose of having different 6 different mattresses what variable is under consideration.
  6. Materials and Methods section: I would suggest to sub divide the section the headings while presenting each test method.
  7. Table 4, kindly discuss the most prominent results and relate it with the thickness or density of the mattress.
  8. Discuss how variation in thickens or density is affecting the different properties.
  9. Kindly compare the results (Numeric values) with the literature and present a comparison among them.
  10. Conclusions, kindly present some of the prominent results with the numeric values.

Author Response

Thank you for the review activity and we are grateful for the opportunity to improve our activity because each observation was another answered question for further analysis and a learning opportunity.

We hope me managed to answer to all the addressed issues.

Best regards,

Authors.

Reviewer 2 Report

This paper presented the thermal insulation capabilities of textile waste and recycled plastic waste integrated with natural fibers. The authors first provided a comprehensive review of the state-of-the-art, then the materials and methods employed in this paper were addressed. The results include moisture absorption, microscopic images and analysis of the fibers, mechanical testing results,  etc. The reviewer suggests a revision of this paper before accepting for publication. Please see the comments below:

  1. Can the authors provide more information of their materials? In the current draft, the reviewer can not find out what materials were tested and reported. What were the weight concentrations of each component?
  2. Since the title of this paper indicates the paper should be about thermal insulation, the reviewer expects to see the majority of results reported in this paper should be thermal-related data and discoveries. However, the current paper draft has included few thermal testing results. This is the major concern of this reviewer. The authors may consider a new title of this paper. 

Author Response

(The authors gave the same response as above.)

Round 2

Reviewer 1 Report

Dear Authors, 

Kindly provide the source of thermal conductivities presented in Figure 1. As these results are not from authors own study, therefore it is recommended to cite the relevant source of information presented in Figure 1. 

Author Response

Thank you for the review activity and we are grateful for the opportunity to improve our activity because each observation was another answered question for further analysis and a learning opportunity.

Reviewer 2 Report

The authors have followed the reviewer's comments and revised the paper. The current paper is well prepared. The reviewer suggest publishing this paper. 

Author Response

(The authors gave the same response as above.)
